25

# Hierarchical sedimentary architecture governs basin-scale solute dispersion: From pre-asymptotic dynamics to uncertainty propagation

- Wanli Ren¹, Yue Fan², Anwen Pan¹, Heng Dai¹\*, Jing Yang³, Mohamad Reza Soltanian⁴, Zhenxue Dai⁵, Songhu Yuan¹
  - <sup>1</sup> State Key Laboratory of Geomicrobiology and Environmental Changes, China University of Geosciences, Wuhan, Hubei, 430074, China.
- 10 <sup>2</sup> Key Laboratory of Geotechnical Mechanics and Engineering of Ministry of Water Resources, Changjiang River Scientific Research Institute, Wuhan, Hubei, 430010, China.
  - <sup>3</sup> School of Land Engineering, Chang'an University, Xi'an, Shanxi, 710064, China.
  - <sup>4</sup>Departments of Geosciences and Environmental Engineering, University of Cincinnati, Cincinnati, Ohio, 45221, USA.
- <sup>5</sup> College of Environmental and Municipal Engineering, Qingdao University of Technology, Qingdao, Shandong, 266033,
   China.
  - <sup>6</sup> College of Construction Engineering, Jilin University, Changchun, Jilin, 130061, China.

Correspondence to: Heng Dai (daiheng@cug.edu.cn)

Abstract. Real aquifers are structured as hierarchical sedimentary systems, where multi-scale heterogeneity and geometric connectivity jointly govern groundwater flow and solute migration. Although the general influence of heterogeneity has been extensively investigated, the scale-dependent effects of hierarchical organization, particularly under basin-scale flow conditions, remain inadequately quantified. In this study, we reconstructed a series of three-dimensional heterogeneous sedimentary architectures at the basin scale and performed numerical simulations to explore dispersion behavior. The results reveal that the geometry and connectivity of dominant lithofacies at macroform scales control macro-dispersion, while finer-scale heterogeneity has only a secondary influence on plume evolution. Furthermore, the evolution of macro-dispersion is characterized by a prolonged pre-asymptotic phase, far exceeding that observed at classical sites such as Borden, indicating that basin-scale solute transport remains non-ergodic over extended times and distances. Uncertainty analysis further identifies a distinct buffering effect inherent to basin systems, in which the aggregation of numerous flow pathways dampens realization-to-realization variability caused by local heterogeneity. When integrated with previously reported laboratory- and

sandbox-scale results from the same site, these findings establish a mechanistic and transferable framework linking hierarchical sedimentary architecture to multi-scale dispersion and uncertainty. This framework advances theoretical understanding of non-Fickian transport and provides practical guidance for large-scale modeling and groundwater management in data-limited regions.

## 1. Introduction

45

The study of solute transport in heterogeneous porous media remains a central challenge in hydrogeology, with significant implications for groundwater science and engineering (Faroughi et al., 2023; Hansen and Berkowitz, 2020; Lee et al., 2018). Decades of field investigations, including the Borden, Cape Cod, MADE and Hanford tracer tests, and laboratory experiments have consistently demonstrated that spatial heterogeneity strongly governs plume migration, leading to non-Fickian dispersion, anomalous breakthrough behaviors, and scale-dependent transport parameters (Sudicky, 1986; Garabedian et al., 1991; Boggs et al., 1992; Berkowitz et al., 2006; Botter et al., 2008; Chen et al., 2010, 2012). A key unresolved question is how quantifiable hierarchical sedimentary architecture, spanning from facies-scale organization to basin-scale structure, governs macroscopic spreading and dispersion across scales (Dai et al., 2005; Ramanathan et al., 2010; Soltanian et al., 2015a).

While stochastic theories and numerical upscaling frameworks have provided valuable tools to connect flow and mixing processes across spatial and temporal domains, many studies continue to compress heterogeneity into effective parameters (Dagan, 1984; Gelhar, 1992; Dentz et al., 2011). Such simplification obscures the mechanistic role of multi-scale sedimentary architecture in shaping flow pathways and controlling plume dynamics, and can bias spreading and mixing-relevant predictions (Fitts, 1996; Neuman and Tartakovsky, 2009; Dentz et al. 2023; Lester et al. 2016; Yin et al., 2023). Scheibe and Freyberg (1995) were pioneers in systematically introducing the pre-existing concept of hierarchical organization from sedimentology into hydrogeology to characterize aquifer heterogeneity laying the foundation for subsequent multi-scale modeling frameworks.

By characterizing the spatial organization of lithofacies across multiple scales, hierarchical sedimentary architecture highlights that aquifer heterogeneity is not a single-scale problem but rather results from the nested and superimposed geological units at different hierarchical levels (e.g., micro-, meso-, and macro-facies) (Dai, 2004; Soltanian and Ritzi, 2014). This hierarchical framework systematically integrates the multi-scale heterogeneities into a unified representation, quantitatively linking sedimentary attributes (such as lithofacies volume proportion, mean length, and statistics of hydraulic conductivity) to transport metrics (such as dispersion and mixing). Lagrangian-based theoretical models derived from this framework have successfully explain the control mechanism of scale dependent transport which is observed in laboratory and site-scale experiments (Soltanian et al., 2015b; Dai et al., 2020; Jia et al., 2023; Ma et al., 2025). Recent work further emphasizes that the role of lithofacies geometric attributes, particularly mean length and connectivity, control solute dispersion and mixing in alluvial systems (Ramanathan et al., 2008; Soltanian et al., 2020; Ershadnia et al., 2021; Soltanian et al., 2017). High-resolution numerical simulations provide more intuitive and detailed information on fluid flow trajectory

65

and interfacial reaction processes, thereby validating and supplementing the assumptions of the Lagrangian theoretical models (Ren et al., 2022).

However, solute transport at the regional scale or basin-scale with long travel distances and laterally extensive pathways still lacks a quantitative characterization constrained by hierarchical sedimentary architecture. Most validations have been limited to laboratory or site-scale studies. Over the past decade, kilometer- to basin-scale investigations have also begun to transition from simplified effective-parameter descriptions toward high-resolution, architecture-resolved analyses. Architecture attributes and conductivity (*K*) statistics has been shown to organize macroscopic dispersion and non-Fickian behavior: high-resolution and architecture-resolved models built in the Lagas groundwater subbasin link lithofacies geometry and connectivity to macro-dispersion and fast pathways/tailing contrasts (Carle et al., 2006); kilometer-scale simulations (Pauloo et al., 2021) demonstrate that aquifer anisotropy and seasonal recharge/pumping-driven shifts in mean flow direction modulate the non-Fickian tails; and regional upscaling research under transient boundaries (Guo et al., 2019) shows that late-time tails cannot be reproduced by multirate mass transfer (MRMT) model calibrated under steady flow because mass transfer between high- and low-*K* units varies with boundary-driven internal gradients. Even with these advances, transport at regional/basin scale remains under-quantified in an architecture-resolved sense.

Accordingly, this study integrates hierarchical sedimentary structures characterization, numerical simulations, uncertainty analysis, using the Qiqihar aquifer system as a basin-scale testbed. Our objectives are to (i) quantify the relative contributions of sedimentary architectural attributes, hydraulic statistics, and source size under field-representative aquifer and large-scale flow fields, (ii) identify which hierarchical scales of heterogeneity dominate solute dispersion, and (iii) evaluate how structural (model-form) or parametric/data uncertainty propagate to transport predictions across scales. This work aims to provide guidance for more reliable predictions of solute transport in field-representative aquifer systems.

The paper is organized as follows: Part II the geographic background of the study area, borehole data, and sedimentary architecture analysis methods; Part III describes the construction of the multi-scale heterogeneous structural model and the simulation process of solute transport; Part IV shows the simulation results and conducts uncertainty analysis to explore the influence of sedimentary architecture and permeability parameters on solute dispersion; Part V summarizes the main conclusions and puts forward the future research.

## 2. Method

# 2.1 Overview of the study area

The study area is located on the east bank of the Nen River in Qiqihar City, Heilongjiang Province, China (Figure 1a). The Nen River defines the western boundary of the study area, and flows from the northeast to the southwest, with a total length of about 22 km and a width ranging from 400 m to 900 m. The regional terrain is gently inclined, sloping from north to south at less than 

concentrated in the phreatic aquifer. The present study therefore investigates solute transport processes within the phreatic aquifer to clarify the mechanisms of solute migration under the influence of sedimentary architecture.

Figure 1. (a) Geographic location map of the study area; (b) Boreholes and cross-sections

A total of 57 boreholes (see in Figure 1b) were collected for this study. According to the borehole data, the phreatic aquifer is composed of thick Quaternary unconsolidated deposits. The lithofacies of the aquifer is dominated by sandy gravel,






gravelly medium- to coarse-grained sand, and medium- to fine-grained sand. The aquifer thickness varies from 22.1 m to 56.9 m, and the depth to the water table varies between 1.75 m and 5.86 m.

The phreatic aquifer is primarily recharged by infiltration from the Nen River, as river stage is consistently higher than the groundwater level. Other recharge includes precipitation infiltration and by lateral inflow from the northern and northeastern boundaries. A relatively continuous aquitard, mainly composed of silty clay and clay, is located below this aquifer. However, it is locally absent near the Nen River and in the southern part of the study area, forming hydraulic windows that allow vertical connectivity between the two aquifer systems.

At the regional scale (kilometers), the phreatic aquifer exhibits a strongly layer-controlled, multi-scale heterogeneity. Along the principal groundwater flow direction from upstream to downstream, the dominant lithofacies progressively transition from gravelly/medium-coarse sand to medium-fine sand. Away from the Nen River and into the floodplain, sediments become finer overall, with local development of low-permeability clay—silt lenses Vertically, a pervasive "coarse-over-fine" organization is observed: the upper section is dominated by gravel and gravelly coarse/medium sand with moderate sorting, whereas the lower section consists mainly of medium-fine to fine sand with better sorting.

## 2.2 Parameter determination

## 2.2.1 Sediments heterogeneous parameter

Eight cross-sections (Figure 1b), oriented parallel to (defined as *x*-direction, sections 1-4) and perpendicular to (defined as *y*-direction, sections 5-8) the regional groundwater flow, were used to constrain the sedimentary structure characteristics. Following the multi-scale hierarchical framework proposed by Dai et al. (2004) and Ritzi and Allen King (2007), this study divided the lithofacies types of the aquifer into two hierarchical scales (Figure 2). At Scale I, eight mutually exclusive lithofacies were defined based on permeability contrasts: gravel (G), coarse sand (CS), medium-coarse sand (MCS), medium sand (MS), medium-fine sand (MFS), fine sand (FS), sub-sandy loam (SL), and clayey loam (C). At Scale II, these were aggregated into three composite units: gravel coarse sand (GCS), medium sand (MFS) and sandy clay (SC). Detailed delineations of the lithofacies at Scale I and Scale II are provided in Figures S1-S8 and S9-S16 in the Supplementary materials, respectively.



Figure 2. Schematic diagram of lithofacies composition of the layered structural model of Qiqihar phreatic water aquifer

The heterogeneous architecture models were produced via conditional stochastic simulation method under a transition-probability Markov-chain scheme, constrained by 57 boreholes and parameterized by lithofacies volume proportions and mean lengths. To bias-correct parameter estimates affected by incomplete exposure of sections, a Bayesian updating scheme (Dai et al., 2005; White and Willis, 2000) and pixel-based image analysis were employed to calculate lithofacies lengths and volume proportions at both hierarchical scales. Comparison of the volume proportions computed from whole-sections ( $P_S$ ) with those calculated from boreholes along the sections ( $P_D$ ) revealed a close agreement (see Table S1 in Supplementary materials), indicating that these sections are sufficiently representative for estimating structural parameters.

The lithofacies volume proportions (P) and mean vertical thicknesses ( $L_Z$ ) were derived from discrete borehole data, whereas mean horizontal lengths along x- and y- directions ( $L_X$ ,  $L_Y$ ) were obtained from the cross-sections. Statistics of heterogeneous parameters are listed in Table 1. Define the horizontal anisotropy coefficient as  $\alpha = L_Y / L_X$ , and the vertical anisotropy coefficient as  $\varepsilon = L_Z / [(L_X + L_Y) / 2]$ . According to Table 1, most lithofacies are horizontally isotropic at both scales, however, the MCS and SL lithofacies extend longer in the x- direction and the C lithofacies extend longer in the y-direction at Scale I. The mean anisotropy ratio decreases from  $\sim$ 0.005 (Scale I) to  $\sim$ 0.003 (Scale II), evidencing reduced apparent heterogeneity with model coarsening.

Table 1. Statistics of heterogeneous architecture parameters at Scale I and Scale II

| Scale   | Lithofacies | Р    | $L_X(m)$ | $L_Y(m)$ | $L_Z(m)$ | а     | 3     |
|---------|-------------|------|----------|----------|----------|-------|-------|
| Scale I | G           | 0.34 | 2331.33  | 2495.40  | 8.94     | 1.070 | 0.004 |
|         | CS          | 0.06 | 2300.95  | 2310.10  | 11.41    | 1.004 | 0.005 |
|         | MCS         | 0.11 | 874.44   | 1187.37  | 9.43     | 1.358 | 0.009 |
|         | MS          | 0.13 | 1525.90  | 1496.85  | 6.45     | 0.981 | 0.004 |

|          | MFS | 0.17 | 1516.81 | 1801.83 | 8.26 | 1.188 | 0.005 |
|----------|-----|------|---------|---------|------|-------|-------|
|          | FS  | 0.09 | 881.78  | 1009.77 | 3.99 | 1.145 | 0.004 |
|          | SL  | 0.04 | 640.32  | 791.79  | 2.55 | 1.236 | 0.004 |
|          | С   | 0.07 | 1098.27 | 763.32  | 2.48 | 0.695 | 0.003 |
|          | GCS | 0.50 | 2205.99 | 2118.44 | 7.35 | 0.960 | 0.003 |
| Scale II | MFS | 0.39 | 1531.79 | 1412.87 | 5.06 | 0.922 | 0.003 |
|          | SC  | 0.11 | 817.75  | 842.73  | 2.60 | 1.030 | 0.003 |

## 2.2.2 Hydraulic conductivity



Hydraulic conductivity (*K*) was constrained using a combination of pumping-test analyses and grain-size-based empirical equations. A total of 45 pumping experiments were collected in the study area (including steady flow pumping tests and unsteady flow pumping tests) and the corresponding *K* values were obtained using analytical solutions and type-curve matching. However, the resulting *K* values are most representative for coarse-grained media. For fine-grained sediments, *K* was estimated from empirical equations. The soil samples collected from the shallow part of the study area were first analyzed for grain-size distribution, and *K* values were then estimated using the empirical equations summarized by Vuković and Soro (1992):

$$K = \frac{g}{v} C\varphi(n) d_e^2 \tag{1}$$

Where: K is the conductivity (m/d); v is the kinematic viscosity (m<sup>2</sup>/s); g is the gravity, taking the value of 9.81 m/s<sup>2</sup>; C is the empirical coefficient (provided in Table S2 in the Supplementary materials);  $\varphi(n)$  is the dimensionless porosity function, in which  $n=0.225\times(1+0.83)^n$ ,  $\eta=d_{60}/d_{10}$ , usually referred to as coefficient uniformity;  $d_e$  is the particle size (mm) in the cumulative distribution curve of the media particle size when the weight of the sample is e%.

Grain-size analyses showed that all samples had characteristic diameters of less than 0.2 mm (data from Dai et al., 2022). By comparing the K results calculated by various empirical formulas, the USBR method was finally adopted for the lithofacies SL. For the very low-permeability C unit, where grain-size methods are less reliable, K was inferred from a regional Plasticity Index-K (PI-K) empirical relationship, where the PI data was collected from clayey soils in the Qiqihar area. Through the above methods, the statistical results of the mean and variance of the logarithmic conductivity ( $\mathcal{Z}$ =ln(K)) of each lithofacies on two scales were obtained, as shown in Table 2.

Table 2. Univariate statistics of conductivity for different hierarchical lithofacies at different scales

| Scale   | Lithofacies | $\overline{\Xi}(\mathrm{m/d})$ | $\sigma_{\scriptscriptstyle{\Xi}}^2$ |  |
|---------|-------------|--------------------------------|--------------------------------------|--|
| Scale I | G           | 4.163                          | 0.019                                |  |
|         | CS          | 3.684                          | 0.059                                |  |



|          | MCS | 3.165  | 0.126 |
|----------|-----|--------|-------|
|          | MS  | 2.775  | 0.177 |
|          | MFS | 2.280  | 0.096 |
|          | FS  | 0.868  | 0.321 |
|          | SL  | -0.251 | 0.116 |
|          | C   | -4.212 | 0.118 |
| Scale II | GCS | 3.829  | 0.080 |
|          | MFS | 2.336  | 0.402 |
|          | SC  | -2.105 | 0.689 |

## 2.3 Construction of a heterogeneous structural model

Taking into account the aquifer structure and the topographic and geomorphological features of the study area, the simulation domain is set to be about 20 km in the *x*-direction, 22 km in the *y*-direction, and 50 m in the *z*-direction. Note that the purpose of this study is not intend to accurately simulate groundwater dynamics or pollution plumes in the study area, but rather to discuss the influence of sedimentary structure and related parameters on solute dispersion. Therefore, for the convenience of modeling and parameter setting, the simulation area was simplified into an approximate square. As listed in Table 1, the SL facies type has the smallest mean horizontal extension (640.32 m), while the MS facies type has the smallest mean vertical extension (0.98 m). To capture lithological heterogeneity in detail, the grid cells were sized at 100 m × 100 m × 0.25 m, resulting in a total of 200 × 220 × 200 cells.

Indicated kriging method based on Markov chain transfer probabilities was used to complete the modeling of heterogeneous architectures. Following Proce et al. (2004) and Ren et al. (2022), the sedimentary architecture was modeled in two stages. First, 50 realizations at Scale II and 50 realizations at Scale I were generated based on proportion and length statistics, respectively. Second, Scale I facies were then mapped onto their corresponding Scale-II domains to yield hierarchical models (hereafter referred to as the multiscale models). The resulting multiscale models thus embed fine-scale lithofacies heterogeneity within a framework that retains large-scale sedimentary architecture, enabling hierarchical coupling across scales. As an example, the three-dimensional and two-dimensional cross-sections of a particular heterogeneous model are shown in Figure 3. In all heterogeneous-architecture models, x = 0 is adjacent to the Nen River; larger x indicates greater distance from the river. Larger y denotes upstream, and smaller y denotes downstream.

Figure 3. An example of multiscale model: (a) 3-D facies, (b) 2-D sections; and Scale II model: (c) 3-D facies, (d) 2-D sections. Section lines at x = 600m, 3000m, 7600m, 11000 m and y = 8000m, 12000m, 16000m, 18000m

Facies in the simulated 2-D sections closely match field patterns and exhibit pronounced stratification. Along the groundwater flow direction, dominant facies transition from G and MCS to MS and MFS. Vertically, the succession fines downward, with coarser textures near the top and finer ones at the bottom.

# 2.4 Flow and solute transport simulation

#### 2.4.1 Hydrogeologic conceptual model



Long-term groundwater level observations indicate that water levels have remained relatively stable over the years, suggesting a stable flow regime. According to multi-year groundwater level records, specified-head boundaries at y = 0 m and y = 22 km (corresponding to the upstream and downstream boundaries) with multi-year mean heads of 146 m and 140 m were set, respectively (Figure 4). The Nen River in the western part of the study area was also generalized as a steady head boundary (144 m). The eastern boundary was specified as a flux boundary, with flux values calculated from the natural hydraulic gradient. The bottom of the phreatic aquifer is an aquitard with overflow discharge, so it was set as the flux boundary. The top boundary is the phreatic surface where recharge is primarily from precipitation, while discharge occurs through evaporation and artificial pumping.


Figure 4. Conceptual hydrogeologic model of the study area and schematic diagram of solute plume planar source release conditions

#### 2.4.2 Groundwater flow simulation

In this study, MODFLOW-2005 (Harbaugh, 2005) was used to calculate the groundwater transport process in saturated porous media. The governing equation for 3-D steady flow is:

$$\frac{\partial}{\partial_{x}} \left( K_{xx} \frac{\partial H}{\partial x} \right) + \frac{\partial}{\partial_{y}} \left( K_{yy} \frac{\partial H}{\partial y} \right) + \frac{\partial}{\partial_{z}} \left( K_{zz} \frac{\partial H}{\partial z} \right) + \omega = 0$$
(2)

where: H is the head [L],  $K_{xx}$ ,  $K_{yy}$ , and  $K_{zz}$  are the principal components of the conductivity tensor [L/T] in the x-, y-, and z directions;  $\omega$  is the source-sink term (1/T), which indicates the amount of water flowing into or out of the aquifer per unit time from a unit volume of the aquifer.

After constructing the heterogeneous sedimentary architecture, stochastic K fields were generated by assigning facies-conditioned K values to each cell based on the spatial distribution characteristics and statistics ( $\overline{\Xi}$  and  $\sigma_{\Xi}^2$ ). Groundwater-flow simulations were run on the same grid as the architecture, ensuring a one-to-one correspondence between facies and K. Based on the recharge conditions of phreatic water, the lithology and thickness of the vadose-zone, city land and farmland, four water-balance subregions have been delineated (shown in Figure 1). Detailed water-balance equations and parameter values are provided in the source and sink calculation section of the Supplementary materials.

## 2.4.3 Solute transport simulation

The random walk particle tracer model program RWHet (LaBolle et al., 1996) was used to simulate the solute transport process. This study focuses on the transport characteristics of nonreactive solutes, and the governing equation for solute transport in saturated porous media defined as:

$$\frac{\partial}{\partial t} [\Theta(\mathbf{x}, t)c(\mathbf{x}, t)] = -\sum_{i=1}^{3} \frac{\partial}{\partial x_{i}} [v_{i}(\mathbf{x}, t)\Theta(\mathbf{x}, t)c(\mathbf{x}, t)] + \sum_{i,j=1}^{3} \frac{\partial}{\partial x_{i}} [D_{ij}(\mathbf{x}, t)\Theta(\mathbf{x}, t) \frac{\partial c(\mathbf{x}, t)}{\partial x_{j}}] + \sum_{k} q_{k}(\mathbf{x}, t)c_{k}(\mathbf{x}, t)\delta_{k}(\mathbf{x} - \mathbf{x}_{k})$$
(3)

where c [M/L<sup>3</sup>] is the dissolved resident concentration; v [L/T] is the velocity;  $\Theta$  [L<sup>3</sup>/L<sup>3</sup>] is the porosity;  $x_{i,j}$  [L] is the distance along the respective Cartesian coordinate axis;  $c_k$  [M/L<sup>3</sup>] is the aqueous phase concentration in the flux  $q_k$  [L<sup>3</sup>/T] of water at  $x_k$ ; and  $\delta_k$  is a Dirac function.  $D_{ij}$  [L<sup>2</sup>/T] is the local-scale dispersion tensor.

In this study, Θ is assumed to be stably isotropic and takes the value of 0.35. The influence of local scale dispersion and molecular diffusion coefficient were not considered in this study, and therefore the corresponding dispersion and diffusion coefficients were taken as zero.

Solute transport was simulated under two distinct source scenarios: a point source and a planar source. Using the bottom of the aquifer near the Nen River upstream as the origin of the coordinate system, the planar source (shown in Figure 4), oriented perpendicular to the groundwater flow, was centered at (10000, 3000, 25) m and extended 14000 m in the x-direction, 100 m in the y-direction, and 50 m in the z-direction. The point source was centered at (10000, 3000, 44) m, and extended 200 m in the x-direction, 200 m in the y-direction, and 2.5 m in the z-direction. A continuous NaCl source with a concentration of 800 mg/L was imposed based on groundwater samples, with a background concentration of zero. Absorption type boundaries were defined at y = 0 m and y = 22 km to allow particles exit the simulation domain at the inlet and outlet boundaries, while reflection boundaries were used for all other boundaries to ensure particles remained within the domain. The solute transport simulation was performed over a period of 10000 days, utilizing the same spatial discretization as the flow model.

The solute transport was measured by the solute concentration moments. According to the definitions of Freyberg (1986), the first moment, normalized by the mass in the solution, represents the position of the solute plume, expressed as the centroid coordinates ( $x_c$ ,  $y_c$ ,  $z_c$ ). The second spatial moment quantifies solute spreading around the centroid, given by the variances ( $\sigma_{xx}^2$ ,  $\sigma_{yy}^2$ ,  $\sigma_{zz}^2$ ). In the subsequent analysis, only the longitudinal component along the groundwater flow direction (y-direction) was retained.

## 3. Results




#### 3.1 Numerical simulation results

# 240 3.1.1 Water flow simulation results

Model validation was performed using groundwater levels measured at 15 observation wells in 2020, whose spatial distribution is shown in Figure 5a. The average simulated heads from 50 realizations were compared with the observed values to evaluate model performance.

Figure 5. (a) Distribution of groundwater level measurement points in 2020; (b) Fitting diagram between simulated water level and measured data

The simulated water levels show good agreement with the observed values, closely following the 1:1 line, which proves that the water flow model constructed in this study reliably captures the groundwater dynamics of the study area. This agreement further indicates that the hydraulic conductivity values derived from different methods are reasonable and consistent.

# 3.1.2 Solute transport simulation results



In stochastic transport modeling, aquifer heterogeneity is represented as independent random field with prescribed statistics. Consequently, transport characteristics are described using ensemble statistics over multiple realizations. Two complementary metrics are commonly employed to quantify macro-dispersion (Dentz et al., 2000a, 2000b), which differ by their averaging procedure: the effective dispersion coefficient ( $D^{eff}$ ) and the ensemble dispersion coefficient ( $D^{ens}$ ). The former is computed by first calculating the moment-based dispersion for each realization and then averaging across realizations. In contrast,  $D^{ens}$  is derived directly from the spatial moments of the ensemble-averaged solute concentration




field. While both metrics characterize macro-dispersion,  $D^{eff}$  more accurately represents the mean spreading in a typical field-scale realization and  $D^{ens}$  includes an "artificial" component from the spatial wandering of plume centroids between different realizations. Under ergodic conditions or at sufficiently long times, these two measures are expected to converge. For interpretation, we also report the corresponding macro-dispersivities,  $\alpha_{eff}$  and  $\alpha_{ens}$ , obtained by normalizing dispersion by the average groundwater velocity.

Figure 6 shows the temporal evolution of plume center of mass y(t) and macro-dispersivity  $\alpha(t)$  for multiscale (red) and Scale II (blue) models under point- and planar-source release conditions. Lines denote ensemble means, and shaded envelopes depict 10–90% confidence intervals from 50 stochastic realizations. Convergence tests indicate that 50 realizations are sufficient to achieve stable statistics. The close agreement between multi-scale and Scale II models, especially under planar-source release (Figure 6c and 6d), demonstrates that at the basin scale, dispersion characteristics can be effectively captured by accurately representing the geometry of controlling lithofacies at larger scales, with diminished influence from smaller-scale heterogeneity. This finding is consistent with the findings of Ramanathan et al. (2010) and Soltanian et al. (2015b), who demonstrated similar behavior using Lagrangian-based macro-dispersion models. Comparable patterns were also observed in the site-scale simulations of Ren et al. (2022) at the Borden site, where dispersivity exhibited a rapid increase and quickly converged to an asymptotic value.

Figure 6. Results of solute concentration center-of-mass transport distances (a, c) and macrodispersivity (b, d) on two scales with time under two release conditions

The time derivative of the mass transport distance corresponds to the mean velocity of solute migration in a given direction. According to Figure 6a and 6c, the average transport velocity of the solute plume is approximately 0.058 m/d for a point-source release and 0.027 m/d for a planar-source release. Using the mean hydraulic conductivity of 12.05 m/d, a regional hydraulic gradient of ~1‰, and an effective porosity of ~0.35 (Table 2), the Darcy-based estimate of the mean







groundwater velocity is 0.03 m/d. The consistency between this calculated velocity and the simulated plume migration further suggests that the model adequately captures the macroscopic solute transport characteristics in the study area. Comparisons with the Borden site highlight distinct timescales: dispersivity at Borden increased rapidly and reached its asymptotic value within a relatively short time, whereas in this study dispersivity approached its asymptote more gradually. This difference likely reflects the higher mean velocity at the Borden site (~0.091 m/d), which accelerates stabilization.

At the release location, high proportions of gravel and medium-to-coarse sand lithofacies lead to strong local-scale heterogeneity in the *K* field. Under point source release scenario, the plume initially samples a limited portion of the heterogeneity, local scale flow velocity deviate significantly from average regional groundwater flow velocity, showing a large fluctuation amplitude and significant uncertainty (shaded areas in Figures 6a and 6b). In contrast, planar source covers a broader portion of the heterogeneous medium at early times. This "source-area enlargement effect" smooths local-scale velocity deviations and explains why confidence intervals are much narrower in the planar-source scenarios. Similar trends were reported by Cao et al. (2018) and de Barros (2018), who showed that enlarging the source area/width markedly narrows uncertainty bands and reduces realization-to-realization spread of plume metrics in heterogeneous aquifers. Theoretically, Dagan (2017) further explained why planar, large-area injections approach ergodic sampling of the conductivity field, smoothing local velocity fluctuations and stabilizing large-scale transport statistics.

The ensemble macro-dispersivity ( $\alpha_{ens}$ ) is more commonly used due to its simpler definition. However, it is found that  $\alpha_{ens}$  primarily captures the statistical characteristics across an ensemble of realizations rather than solute transport in a specific realization. Moreover,  $\alpha_{ens}$  often overestimates the solute dispersion under non-ergodic or pre-asymptotic conditions. Figure 6b illustrates this overestimation significantly and shows that the error associated with  $\alpha_{ens}$  increases progressively with transport time for both the multiscale model and the larger-scale model. The persistence of this overestimation until the end of the 10000-day simulation suggests that plume evolution remains largely governed by advection and has not yet reached a dispersion-dominated regime controlled by local-scale dispersion. Once the solute plume samples a sufficiently large domain of the heterogeneous medium, realization variability is substantially reduced and large-scale transport properties converge to the ensemble mean. Accordingly,  $\alpha_{eff}$  tends toward  $\alpha_{ens}$ , consistent with theoretical predictions.

#### 3.2 Uncertainty analysis of solute dispersion

A recent global sensitivity analysis indicated that a small set of geologically interpretable factors, most notably facies volume proportions and in-facies mean hydraulic conductivity, exerts first-order influence on non-reactive solute dispersion across regional to basin scales (Ren et al., 2023). Motivated by these findings, we next explore the individual contribution of each factor to solute dispersion. To ensure a stable and consistent comparison while minimizing noise between realizations, all subsequent simulations were conducted under planar-source releases and with Scale II heterogeneous models. This configuration focuses on the dominant lithofacies architecture, yielding an ensemble-like depiction of field-scale migration. This choice also provides a good chance to effectively narrow uncertainty bands and establish a clear baseline for systematically evaluating the impact of key parameters.




# 3.2.1 Effect of volume proportions

At Scale II, lithofacies are grouped into three types in decreasing order of permeability: GCS, MFS, and SC, with volumetric proportions of 0.503, 0.385 and 0.112, respectively. Two scenarios are designed by changing the lithofacies volume proportions, while holding all other statistics fixed. In Group A, the proportions were set to 0.33, 0.34, and 0.33, approximating an aquifer with equal volumetric fractions of all lithofacies. In Group B, the mixture was skewed toward the low-permeability unit, with volume ratios of 0.2, 0.3, and 0.5, respectively.

Figure 7 shows the simulation results for different scenarios. With the increase of time, the solute transport distance increases proportionally, and the dispersivity shows a power function growth trend. This behavior is consistent with the well-established theory of large-scale dispersion in heterogeneous media, where transport is characterized by an initial non-Fickian regime followed by a transition towards Fickian behavior at late times (Dagan 1989; Neuman and Tartakovsky, 2009).

Figure 7. Results of (a) solute concentration center-of-mass transport distances and (b) macrodispersivity under different lithofacies volume proportion scenarios

As shown in Figure 7, with the increase of the volume proportion of GCS, the center of mass of the solute plume advances more rapidly and the asymptotic value of dispersivity are also increase, as well as the uncertainty of the simulation results. Across the tested scenarios, the equilibrium macro-dispersivity stabilizes around 170 m after approximately 5000 days. High permeability lithofacies type (e.g., GCS) always provides preferential pathways that accelerate plume migration, while low permeability type (e.g., SC) acts as barriers that restrict plume spreading and extent solute residence times. Our







simulation results reveal that even at the basin scale, modifying lithofacies proportions can still reshape the balance between preferential and retarding controls. Both high-*K* and low-*K* lithofacies exert strong controls on plume transport behaviors. These results are consistent with previous theoretical and numerical studies (Fiori et al., 2010; Amooie et al., 2017; Soltanian and Ritzi, 2014; Puyguiraud et al., 2020), which emphasized that large-scale dispersion is governed not by a single conductivity class, but by the combined influence of extreme permeability contrasts.

A notable finding is that even when the proportion of GCS increases, the differences in plume metrics among scenarios remain modest, and the uncertainty bands widen only slightly. This behavior contrasts with local-scale tracer tests such as those at the MADE site, where strong preferential flow channels lead to early arrivals and heavy-tailed breakthrough curves, and where apparent velocities between observation points can vary by orders of magnitude due to meter-scale *K* variability ( Zheng et al., 2011; Bianchi and Pedretti, 2017). In our basin-scale system (20 km × 22 km), however, the plume travels only a few kilometers after 10000 days, indicating that non-ergodic conditions persist for extremely long times. Under such conditions, changes in lithofacies proportions adjust transport characteristics, but the overall plume dynamics are buffered by the immensity of the flow domain. This suggests that, at regional—basin scales, modifications to lithofacies proportions may act as secondary drivers of plume extent relative to other large-scale controls. This is a nuance often obscured in site-scale studies.

Realization-to-realization variability also reveals a clear scale-dependent trend. Increasing the proportion of high-*K* lithofacies enhances solute mobility but simultaneously amplifies realization to realization variability, whereas low-*K* lithofacies reduces both transport velocity and uncertainty, yielding more uniform outcomes. However, uncertainties do not grow indefinitely with conductive lithofacies dominance but instead plateau due to kilometer-scale spatial averaging. Such findings highlight the importance of explicitly accounting for scale-dependent controls when extrapolating transport models from site to regional contexts, especially for risk assessment and large-scale groundwater management.

# 3.2.2 Influence of the mean value of the conductivity

The influence of conductivity on solute dispersion is primarily based on the difference in the mean *K* values among lithofacies. In the Qiqihar site, the mean *K* values are 46.02 m/d, 10.34 m/d, and 0.12 m/d for of GCS, MFS, and SC, respectively, indicating a pronounced permeability contrast. To isolate the role of individual lithofacies, three model groups were designed in which only the mean *K* of a single lithofacies was increased threefold, while the other two remained unchanged. In Group 1, the mean *K* of GCS was raised to 138.06 m/d, with MFS and SC fixed at 10.34 m/d and 0.12 m/d, respectively. In Group 2, the mean *K* of MFS was increased to 31.02 m/d and in Group 3, the mean *K* of SC was increased to 0.36 m/d. In all cases, the variance of *K* was preserved, and the underlying heterogeneous sedimentary architecture remained unchanged. Thus, all solute transport simulations were carried out within the same structural framework. Solute transport simulations for these scenarios, conducted under planar-source release conditions, are presented in Figure 8.



Figure 8. Results of (a) solute concentration center-of-mass transport distances and (b) macrodispersivity under the variations in lithofacies mean hydraulic conductivity

Figures 8a-8c are the concentration distribution center of mass transport distance results, and Figures 8d-8f are the effective dispersivity results. They all show a clear, asymmetric response when the mean *K* is tripled for a single lithofacies while keeping variance and architecture fixed. Increasing the mean *K* of the most permeable lithofacies (GCS) accelerates plume migration, raises the asymptotic dispersivity, and slightly widens the uncertainty bands. By contrast, tripling the mean *K* of the medium-permeability lithofacies (MFS) counterintuitively reduces both migration distance and dispersivity, whereas perturbing the *K* value of SC produces virtually no change. By calculating the global mean *K* of the 50 realizations, it can be obtained that the global mean *K* value changes from 12.05 m/d of Qiqihar site to 22.20 m/d, 19.49m/d and 14.73 m/d after expanding the mean *K* values of the three lithofacies. This suggests that the MFS perturbation should have facilitated solute transport, contrary to the simulated results. As mentioned in section 2.1, the dominant lithofacies progressively transition from gravelly/medium-coarse sand to medium-fine sand from upstream to downstream. Further analysis of the constructed heterogeneous sedimentary architecture model reveals that MFS lithofacies distributed at the solute plume release location. As the conductivity in the model is generated based on lithofacies distribution, which may result in 1 local simulated anomalies.








Temporal variations in dispersivity also highlight the role of individual facies in shaping plume evolution. In the GCS perturbation, dispersivity increases sharply from the onset of solute release, and its asymptotic value remains well above the baseline case, emphasizing the immediate dominance of coarse type connectivity in channeling solute migration. The MFS perturbation shows reduced dispersivity during the early phase, but values gradually converge toward the baseline at later times. This pattern reflects how diminished conductivity contrasts suppress shear-driven macro-dispersion in the pre-asymptotic regime, while spatial averaging at the basin scale eventually restores ensemble-like behavior. These early and late dynamics are consistent with observations of non-Fickian transport, where temporary suppression and delayed convergence are characteristic of strongly heterogeneous aquifers (Fiori and Dagan, 2000; Dentz et al., 2011).

Collectively, the results indicate that macroscale dispersion may not simply controlled by the average conductivity of the system, but by the extent to which variations in facies properties reallocate groundwater fluxes across the underlying connected pathways. Specifically, enhancing GCS strengthens pre-existing preferential pathways, accelerating plume migration. In contrast, increasing the mean *K* of MFS reduces the contrast with GCS, redistributes part of the flux into slower pathways, and thus suppresses early-time dispersion. This complex flux redistribution mechanism effectively suppresses the overall solute migration rate and reduces the high-*K* channel-driven dispersivity, leading to the observed decrease in both plume metrics. For SC, even a threefold increase leaves it far less conductive than the other facies, preventing it from contributing to the connected high-*K* network. Such asymmetric responses are consistent with previous theoretical and numerical studies, which demonstrate that dispersivity at large scales emerges from the interplay between facies contrasts and connectivity, rather than from the mean conductivity values of single facies (Zinn and Harvey, 2003; Soltanian and Ritzi, 2014). All these observations suggest that model calibration that focus only on asymptotic metrics may overlook early-time transport features that are critical for risk assessment and monitoring system design.

# 4. Discussion

Basin-scale simulations indicate that conservative solute dispersion and its uncertainty are principally organized by facies conductivity contrasts and the geometry of the architectural elements at the larger scale, whereas modest perturbations to smaller-scale structure produce comparatively minor changes in transport metrics. This is consistent with the multiscale sedimentary-architecture paradigm and with prior architecture-resolved studies. This study also distinguishes between effective dispersivity ( $\alpha_{eff}$ ) and ensemble dispersivity ( $\alpha_{ens}$ ) and obtain results consistent with site scale studies:  $\alpha_{ens}$  increasingly overestimates the time-varying dispersion, while  $\alpha_{eff}$  measures spreading within a typical realization and approaches a stable value as sampling increases. The kilometer-scale effective longitudinal dispersivity obtained in this study fall within the classical field scale statistics compiled by Gelhar et al. (1992), who found dispersivity typically in the tens to hundreds of meters at similar scales.

A central finding is the scale-dependent nature of plume stabilization. Over a 10000 days simulation period, dispersivity values initially increased slowly and then stabilized. This indicates that after sufficient time, the plume's large-scale transport behavior begins to stabilize, reflecting a condition where the plume has sampled a representative portion of








the large-scale heterogeneity. However, this stabilization occurs over an extremely prolonged period, highlighting that plume evolution remains in a pre-asymptotic, or quasi-ergodic, state for thousands of days at the basin scale. By contrast, this differs significantly from findings at the Borden site (Ren et al., 2022), where dispersivity was shown to reach an asymptotic value much more rapidly.

This study further highlights that even at the basin scale, dispersivity is not a simple constant. Instead, it is closely related to the aquifer's heterogeneous structure and its hydraulic parameters. Scenario analysis reveals that the facies proportion and mean-conductivity perturbations illustrate two complementary mechanisms by which multiscale heterogeneity governs solute dispersion. Changes in facies proportions alter the statistical balance between preferential pathways and retarding domains, whereas perturbations of mean conductivity reshape the contrast among facies types, redistributing fluxes across the connected network. At the basin scale, both mechanisms exert measurable influences on plume dynamics, but the divergence among scenarios remains modest due to kilometer-scale spatial averaging and persistent non-ergodic conditions. Recent global sensitivity analysis across multiscale heterogeneous media shows a robust ranking for non-reactive solute dispersivity: the facies mean *K* is typically the most influential factor, followed by facies volume proportions and facies mean lengths; variance and some correlation scales contribute less. Importantly, when the heterogeneity integral scale reaches 100m to 1000m, the regional hydraulic gradient becomes non-negligible for non-reactive transport. Our basin-scale results are broadly consistent with these rankings.

Independent 3-D tank/column experiments conducted with sediments from the same Nen-Qiqihar setting (Ma et al., 2022, 2025) provide a crucial mechanistic bridge at smaller scales. These experiments showed clear scale dependence: heterogeneous mixtures produced a much steeper growth of longitudinal dispersivity than single facies columns. The best-fit longitudinal dispersivity was on the order of 0.1 m. These laboratory results directly link dispersion growth to velocity contrasts caused by *K* variability, to lithofacies geometry, and to cross-facies transitions. The consistency between the experimental evidence from this place and the conceptual foundation of basin scale models validates our core finding that mechanisms governed by contrast and connectivity are influential at multiple scales.

Similarly, studies at Borden site emphasize the dominant role of larger-scale facies architecture (proportions, mean lengths) in controlling macro-dispersion. However, the simulation result uncertainties exhibit scale-dependent characteristics. Two additional Borden site simulation results insight help contextualize our findings. First, enlarging the initial source alters early-time dispersion and widens uncertainty envelopes during site scale transport; by contrast, our basin scale results show convergence of uncertainty bands at late times, again reflecting domain scale averaging that mutes source geometry effects far downstream. Second, the increased proportion of more permeable lithofacies at the Borden site significantly amplifies spreading and model uncertainty, while the basin scale models exhibit a weaker response, consistent with buffering by long travel distances and multiple superposed pathways. Mechanistic trends noted in other architecture-resolved modeling (Yin et al., 2020) also align with our results. For example, larger longitudinal mean lengths of high-*K* lithofacies strengthen connectivity and the leading edge, whereas vertical elongation can have the opposite effect; sensitivity to initial source size is strongest at early times.







Taken together, the evidence from our study and a review of the literature supports three key points: (i) contrast- and connectivity-driven mechanisms operate across all scales from lab columns to basins; (ii) the apparent strength of these mechanisms increases from lab to field sites but decreases from field sites to basins; and (iii) for regional-scale prediction, it is often sufficient to preserve dominant geological contrasts, lithofacies proportions, and directional mean lengths, while problems near a source or at early times still require finer architectural resolution and explicit treatment of source size. Importantly, this work highlights that reliable predictions of regional solute migration require not only parameterizing mean values but also preserving conductivity contrasts and lithofacies connectivity, as well as accounting for scale-dependent averaging effects. By integrating these insights, our study advances a more comprehensive understanding of dispersion at the watershed scale and provides a conceptual basis for designing monitoring networks and management strategies that are robust under conditions of limited site characterization.

#### 5. Conclusions

This study provides a comprehensive analysis of multi-scale heterogeneity and its influence on non-reactive solute dispersion and modeling uncertainty within the Nen River Basin aquifer system. By integrating a hierarchical architectural model with numerical simulations and cross-scale validation, we have drawn several key conclusions that advance the understanding of transport in basin-scale heterogeneous media.

First, our findings demonstrate that the geometry of the dominant, large-scale lithofacies is sufficient to characterize solute dispersion, with smaller-scale architectural details having a comparatively minor influence. The evolution of dispersivity in our simulations is characterized by a long pre-asymptotic phase, which contrasts sharply with the rapid stabilization observed at sites like Borden. This unique behavior highlights that at the basin scale, transport remains in a non-ergodic state for quite a long period of time, requiring a new perspective on long-term plume dynamics that accounts for this prolonged pre-asymptotic regime.

Second, we clarified the primary sources of modeling uncertainty. Our results show a "buffering effect" inherent to basin-scale systems, where long travel distances and numerous superimposed pathways dampen the variability caused by local heterogeneity. This contrasts with local-scale studies where preferential channels can lead to orders-of-magnitude variability in plume metrics. We found that the geometry of the contaminant source significantly impacts early-time uncertainty, but this effect is muted at later stages by large-scale spatial averaging. This underscores that for basin-scale problems, preserving dominant geological contrasts and connectivity is crucial, while the influence of local details diminishes over time.

Finally, our study provides a critical conceptual basis for future modeling and management strategies. We established a robust, multi-scale framework for analyzing transport in complex aquifers. This work highlights that reliable regional predictions require not only accurate parameterization but also an explicit accounting for scale-dependent averaging effects and the unique behavior of dispersion in vast, non-ergodic systems. This approach is essential for designing effective monitoring networks and robust risk assessment models in areas with limited site characterization.

#### **Author contribution**

Conceptualization: WLR and ZXD. Data curation: WLR. Formal analysis: WLR, AWP and YF. Funding acquisition: WLR and HD. Modelling and software: YF, MRS and JY. Visualization: AWP and WLR. Writing – original draft preparation: AWP. Writing – review & editing: WLR, YF, AWP, HD, JY, MRS, ZXD and SHY. All authors have read and agreed to the published version of the manuscript.

# **Competing Interests**

One of the (co-)authors is a member of the editorial board of Hydrology and Earth System Sciences.

## Acknowledgments

This work is funded by the National Natural Science Foundation of China (42302291, 42422208).

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
