# Peer review of "Hierarchical sedimentary architecture governs basin-scale solute dispersion: From pre-asymptotic dynamics to uncertainty propagation"

_EGUsphere, 2025_

## Referee Comment (RC2)

This study combined borehole soil sampling, groundwater level data, and groundwater flow and transport modeling to examine the impact of subsurface heterogeneity on dispersion at the basin-scale. Overall, the dataset and modeling framework are promising, and the study has the potential to make a valuable contribution once the issues below are addressed.

Major comments:

1. The novelty and knowledge gap can be better articulated. In particular, the current introduction may not set the stage for this study well. Consequently, the three research objectives lack literature review support and strong motivations.

- For instance, the reviewer would be curious what motivated the author to perform an uncertainty analysis. Was it because existing literature shows a wide range of different observations in the field? Was it because a new model was used and the model involved some uncertain parameters whose uncertainty is unknown? Was it because there has been an inconsistency between existing observations and models? Or something else?

- Another example is why did they want to quantify the relative contributions of sedimentary architectural attributes, hydraulic statistics, and source size? There seems to be no well-organized discussions on the existing knowledge about how these three factors control the dispersion processes under field-representative aquifers and large-scale flow fields.

- A similar question arises for the uncertain proportion part.

2. The writing and logical flow can be improved through substantial revisions. Please see the specific comments for some examples.

3. The manuscript involves quite a number of jargons and unclear sentences, which may require clarification or rephrasing. Below are some examples:
- "From pre-asymptotic dynamics to uncertainty propagation"
- "geometric connectivity"
- "general influence"
- "hierarchical organization"
- "Although the general influence of heterogeneity has been extensively investigated, the scale-dependent effects of hierarchical organization, particularly under basin-scale flow conditions, remain inadequately quantified."
- "lithofacies"
- "macroform scales"
- "macro-dispersion"
- "finer-scale heterogeneity"
- "Non-ergodic"
- "basin-scale structure"
- "macroscopic spreading and dispersion across scales"

- "A total of 57 boreholes … were collected"
- "To bias-correct parameter estimates affected by incomplete exposure of sections, …"

Specific comments:

1. What is the relationship between "finer-scale", "micro-scale", "meso-scale", "macro-scale", "macroform scales", "large-scale", "laboratory-scale", "facies-scale" "field-scale", "region-scale", "kilometer-scale", "basin-scale"? Without clear definitions or quantitative descriptions (like "kilometer" did), it is hard to understand the specific scales.

2. Line 30: Has the "laboratory- and sandbox-scale study" been done in this work? If not, this argument can be misleading.

3. Line 30: The reason why the current study can support studying groundwater management at data-limited regions is not trivial? Please briefly explain the reasoning herein or in the main text.

4. Line 60: It is unclear how the arguments "Recent work …" and "High-resolution …" logically connects with each, as well as how they connect with the prior discussions. That said, it is hard to catch the key message behind these arguments.

5. Line 65: There was no discussion on solute transport at other scales previously. It is hard to see the connection between the prior paragraph to this statement.

"Most validations have been limited to laboratory or site-scale studies." cannot be justified by the current introduction. Also, it's unclear what point the authors would like to make.

6. Line 70: How can "kilometer-scale" be considered as "high-resolution"? What resolution can be considered as "high"?

7. Line 75: "late-time tails …" This can be good motivation for this work, while the argument "Even with these advances, ..." makes the gap ambitious.

8. Line 80: As suggested in my first major comments, the current introduction didn't provide a comprehensive literature review and clear research trajectory to motivate the objectives.

9. "Part II the" is missing a verb.

10. Section 2 - Method: Each subsection of the method section includes both methodology(i.e., general approach) and specific model setup (i.e., parameters and data), which sometimes can affect readability. I would suggest separating them into a methodology section and a parameterization (or data collection and model setup) section. I believe this reorganization will substantially improve the clarity and quality of the paper.

11. Line 90: "The Nen River defines …" Why is this matter?

"The regional terrain is gently…" Is there any support from literature or the collected data? if so, I would either cite the references or clarify the data support.

12. Figure 1 caption: More details should be provided about the selections of boreholes and cross-sections, as well as the rational or purposes of such decisions.

Additionally, what were the cross-sections used for? Or what kind of information was collected from the cross-sections?

13. Figure 2: The figure is hard to interpret without providing more details about the meanings of the acronyms and the plotted regions.

In the right figures, how can one interpret the figure? Are they referring to the horizontal or vertical view of the aquifer, or something else?

How are the right figures connected to Figure 1b.

14. Line 130: Can you elaborate what does it mean by "parameterized by lithofacies volume proportions and mean lengths." or provide a brief introduction about the parameterization?

15. Line 130: What is the "bias-correct parameter"? What does it mean by "To bias-correct parameter estimates affected by incomplete exposure of sections,"?

16. Line 135: How were $P\_S$ and $P\_D$ defined?

17. Table 1: It is hard to understand the physical meanings of the parameters and how they were derived. I would suggest providing a schematic or conceptual plot and label them in the plot at least for some of them (e.g., Lx, Ly, Lz), while providing the formula to compute P.

18. Line 155: I guess you were referring to the characteristic diameters of the grains. Please clarify.

What is USBR?

19. Table 2: The unit should be ln(m/d). Is simga dimensionless? If not, the unit should be provided.

20. Line 175: The authors are recommended to justify that 50 realizations are statistically grounded by adding relevant references.

21. Figure 3 provides a more clear explanation of part of the methodology. Based on that, I would suggest using some plots (e.g., a and c) to create a conceptual figure to describe the entire methodology and adding a short subsection to introduce the conceptualization at the very beginning of Section 2 Method. This will make the rest easier to follow.

22. Figure 4. What is the blue rectangle representing? Are the boundary conditions applied only to the lines or the faces?

23. Line 225: I guess planar source was indicated in Figure 4, while point source was not. Additionally, the authors should explain the rationale of simulating the two different sources, regarding the real-world processes they are representing.

24. Line 250: I would suggest reporting the error or correlation between model and observation. Additionally, in addition to the fitness between the averaged simulated water levels and measured ones, reporting the errors that reflect the fitness of simulated water levels for each realization or their statistical values (e.g., mean and standard deviation of the error of each realization) would better verify the robustness of the model results.

25. Line 280: Add references to the Borden site results.

Why does higher mean velocity accelerate stability? Was it due to stronger mechanical dispersion, or something else?

26. Line 305: The sensitivity and uncertainty methods were not introduced in section 2. Please provide the details about the specific methods and the analyzed factors and parameters.

27. Figure 7 caption: Please clarify what each model was representing

28. Line 350: What does scale-dependent tread mean? How can realization-to-realization variability reveal the "scale-dependent tread"? What is the underlying rational and theoretical basis?

29. Line 355: It seems Scale II was used for these analyses. Can you comment on whether the impact of varying K values on flow and solute transport will remain similar between Scale I and Scale II models?

30. Line 415: What is the reason for this difference? What is the implication for characterizations and simulations of contamination transport at different sites?

I will probably talk about the key finding of these studies and then make the comparisons with the Borden site in the second last paragraph of this section given that there were several comparisons (similarity vs difference).

31. Line numbers should be appended to each line for the reviewer to pinpoint their comments.

---

## Author Comment (AC4)

**Response to comments from Anonymous Referee #1**

**Overview of Anonymous Referee #1:**

This paper, based on real basin study area, systematically analyzes the scale effects of solute transport and dispersion in heterogeneous aquifers using a hierarchical sedimentary framework and numerical simulation. Overall, the manuscript has a good topic selection and relatively advanced research methods, aligning with research frontiers in non-Fick solute transport in heterogeneous aquifers. Through detailed results presentation and comprehensive discussion, it provides practical guidance for large-scale modeling and groundwater management in data-limited regions. However, there is still some space for improvement in areas such as writing format, methodological hypotheses, quantification of results, and the applicability of the research.

**Reply:** We sincerely thank the reviewer for the constructive and insightful comments, which helped us improve the manuscript in terms of clarity, methodological transparency, quantification of key findings, and applicability. In the revised manuscript we will address each of his/her observations/suggestions.

**Specific comments:**

(1)In the Abstract and Discussion sections, authors mentioned "…prolonged pre-asymptotic phase, far exceeding that observed at classical sites such as Borden.", it is suggested that quantitative metrics can be provided here to make the difference in solute transport characteristics between sites and basins more intuitive. Furthermore, the authors mentioned "mechanistic and transferable framework" and "practical guidance" at the end of the Introduction section, but these are broad descriptions. A more accurate and intuitive expression would be better, such as which observable parameters are most important for predictable diffusion indices.

**Reply:** Thank you for this comment. We have revised the text of the **Abstract** and **Discussion** sections, added some quantitative metrics and provide more specific expressions. Below are the specific modifications in the tracking version manuscript.

The red content is the modified information, and the black content represents the original text (the same applies below).

In the **Abstract** section: The results reveal that the geometry and connectivity of dominant lithofacies at macroform scales control macrodispersivity, while finer-scale heterogeneity has only a secondary influence on plume evolution. Furthermore, the evolution of macrodispersivity is characterized by a prolonged pre-asymptotic phase, approaching a quasi-steady state after around 5000 days, with an asymptotic stability value of 170m. This timescale is nearly 10 times longer than that inferred from the Boden site, where macrodispersivity approaches the asymptote after approximately 400 days, stabilizing at around 0.4m.

In the **Discussion** section: A central finding is the scale-dependent nature of plume stabilization. Over a 10000 days simulation period, $\alpha_{eff}(t)$ initially increased slowly and then stabilized, but this stabilization occurs over a relatively prolonged period (around 5000 days) , indicating that plume evolution remains in a pre-asymptotic (quasi-ergodic) state for thousands of days at the basin scale. This behaviour contrasts with site-scale results at the Borden site (Ren et al., 2022), where $\alpha_{eff}(t)$ was shown to reach an asymptotic value much more rapidly (after 400 days), plausibly facilitated by higher mean groundwater velocities. In the present basin-scale system, the solute explores only a small fraction of the heterogeneous flow field over a considerable period of time, the effective transport response is "buffered" by the domain immensity. Source size exerts a qualitatively consistent effect across scales: expanding the source promotes broader early-time sampling of heterogeneity and thereby reduces inter-realization variability, yielding narrower uncertainty during transport process. By contrast, the uncertainty band for $\alpha(t)$ tends to stabilize or even decrease later at the basin scale, again reflecting domain scale averaging that mutes source geometry effects far downstream. In the sitescale tracer experiments (e.g., MADE, Borden), meter-scale $K$ variability may lead to apparent velocities differ by orders of magnitude between observation points, and strongly connected preferential pathways commonly yield pronounced early arrivals and heavy-tailed breakthrough curves (Zheng et al., 2011; Bianchi and Pedretti, 2017), thereby increasing predictive uncertainty in plume evolution and arrival-time statistics. Simulations at the Borden site well demonstrate that the increased proportion of more permeable lithofacies significantly amplifies solute dispersion and output uncertainty. Basin-scale models, however, exhibit a weaker response, consistent with the buffering effect of long travel distance and multiple overlapping pathways.

(2)The last part of the Introduction section is the structure of the paper. It is suggested that word "part" can be changed to "section", and that the title of the Section 2 can be changed to "methods". Section 2.2.1 should be renamed to "Sedimentary heterogeneity parameters" or "Sediment heterogeneity parameters."

**Reply:** We agree and have revised the expression accordingly. We have replaced "part" with "section" in the Introduction section, and have renamed Section 2 to "Methods," and revised Section 2.2.1 to "Sediments heterogeneity parameters". Below are the specific modifications in the tracking version manuscript.

Revised information is in Line 103-107 in the tracking version manuscript: The paper is organized as follows: Section 2 introduces the geographic background of the study area, borehole data, and sedimentary architecture analysis methods; Section 3 describes the construction of the multiscale heterogeneous structural model and the simulation process of solute transport; Section 4 shows the simulation results and conducts uncertainty analysis to explore the influence of sedimentary architecture and permeability parameters on solute dispersion; Section 5 summarizes the main conclusions and outlines future research directions.

(3)In the text, background information and existing research should be described using the present or past tense; the Methods section of the paper should use the past

tense or passive voice; the Results/Discussion section should use the present tense. Please make the corresponding modifications throughout the paper.

**Reply:** We have now systematically revised verb tense usage throughout the manuscript.

(4)57 boreholes were used to construct 20 km × 22 km × 50 m models in this study. While this is a common practice in basin scale modeling, its representativeness still needs to be effectively evaluated since this research focuses on the heterogeneity of the aquifer. It is recommended that the authors supplement the relevant content in the manuscript.

**Reply:** Thank you for raising the important issue of data representativeness. In fact, within a 20 km × 22 km area, this study used not only 57 boreholes but also 8 cross-sections as constraints to represent the potential heterogeneity of the aquifer. The primary objective of this study is not to reproduce a specific site plume deterministically, but to quantify how hierarchical sedimentary architecture and associated parameter uncertainties govern basin-scale dispersion under field-representative flow conditions. Consistent with common practice in regional/basin-scale hydrogeological modeling, we therefore treat the borehole data as hard constraints and rely on a hierarchical, transition-probability/Markov-chain–based geostatistical framework to stochastically populate the inter-borehole space and to explicitly quantify geological uncertainty through ensembles of conditional realizations. To more clearly address these considerations, we have added a new paragraph in the end of the Section 2.2.1.

Revised information is in Line 175-183 in the tracking version manuscript: In this study, the 57 boreholes provide hard conditioning data for facies occurrence and aquifer thickness, whereas the eight cross-sections supply additional structural constraints on lateral continuity and stratigraphic organization along and across the principal directions. This study is not aim to deterministically reproduce specific in-situ plumes, but rather quantifies how hierarchical sedimentary architecture and associated parameter uncertainties govern basin-scale dispersion under field-representative flow conditions. The resulting heterogeneous model was intended to be statistically representative, while local connectivity in data-poor areas was treated as uncertain and

quantified through a set of conditionally realizations. Although boreholes and corresponding cross-sections are more densely packed in the central and western parts of the study area and relatively sparse in the eastern area, this is sufficient to serve the objectives of this study.

(5)A constant porosity of 0.35 is not sufficiently for facies ranging from gravel to clay, but this simplification is acceptable if the research objective is solely to resolve the control of the K-field and architecture on dispersions. It is recommended that the assumptions for this parameter be explicitly stated in Section 2.4.3, and that any potential biases introduced by these assumptions be briefly discussed in the discussion section. Similarly, setting $D_{ij}$=0 implies that macrodispersion is only caused by non-uniform velocity fields. This is reasonable for studying pre-asymptotic behavior of structural controls, but it might underestimate dispersion compared to real systems. It is recommended to add discussion of this aspect in the parameter settings and Discussion section.

**Reply:** Thank you for this comment. We have explicitly stated in Section 2.4.3 that the simplification of setting a constant porosity and neglecting the effects of local-scale dispersion and molecular diffusion is reasonable to identify the structural and *K*-field controls on macrodispersion. Setting $D_{ij} = 0$ implies that solute dispersion arises solely from non-uniform velocity fields (macrodispersion driven by heterogeneity/architecture), which is appropriate for diagnosing pre-asymptotic structural controls but could underestimate total dispersion relative to real aquifers where local dispersivity and molecular diffusion contribute. The limitations and applicability boundaries have also been discussed in both the Section 2.4 and the Discussion Section.

Revised information is in Line 261-267 in the tracking version manuscript: In this study, Θ was assumed to be stably isotropic and was set to the value of 0.35. Although setting a constant porosity may lead to deviations in the time required to reach a certain stage and the absolute value of the dispersion index plotted on the time axis, this study, however, emphasized the influence of aquifer structure and *K*-statistics

under consistent settings, where the spatial heterogeneity and connectivity of the corresponding velocity field were not significantly determined by subtle spatial variations in porosity. Another advantage of this choice is to avoid introducing other poorly constrained parameter fields into the model. For the same reason, the influence of local scale dispersion and molecular diffusion coefficient was not considered in this study, and therefore the corresponding dispersion and diffusion coefficients were taken as zero.

Revised information is in Line 504-508 in the tracking version manuscript: It must be acknowledged that neglecting porosity variations and molecular diffusion processes in this study may lead to an underestimation of early plume smoothing and lateral mixing, potentially delaying a significant convergence to Fick behaviour. However, at the basin scale and in the long-distance travel considered in this paper, structure-controlled velocity variations are expected to dominate the dispersion index; therefore, the main conclusions regarding relative lithofacies proportions and connectivity remain unchanged.

(6)The existing Uncertainty analysis section is a "scenario analysis" What is the basis for setting up the comparison groups for volume proportion and conductivity? In other words, is this reasonable in terms of geological conditions? Please provide further explanation or emphasize the application scenario of this setting to enhance the guiding significance of the conclusions.

**Reply:** Thank you for this comment. At the beginning of Section 3.2, we clarify that the uncertainty analysis is implemented as an scenario analysis. The scenario design was motivated by a published global sensitivity analysis indicating that facies volume proportions and in-facies mean hydraulic conductivity exert first-order influence on non-reactive solute dispersion across regional to basin scales. From a geological perspective, in fluvial–alluvial systems the relative abundance of channel-belt coarse deposits (e.g., gravel/sand bodies) versus floodplain fine deposits can vary substantially in planar terms, reflecting differences in depositional energy, channel migration/avulsion style, floodplain development, and base-level conditions. We

therefore use proportional end members to represent plausible depositional settings. Specifically, Group A (near-equal proportions) represents a more mixed and interbedded architecture consistent with frequent channel migration and facies switching, whereas Group B (fine-dominated mixtures) represents a low-energy and/or distal floodplain setting where fine deposits are more prevalent and coarse bodies are more isolated. The mean-$K$ perturbation scenarios keep the architecture fixed and any changes in dispersion can be attributed to altered inter-facies conductivity contrast and the resulting redistribution of flow among facies (K×3 corresponds to a medium to high level of hydraulic-property uncertainty). As is well known, hydraulic conductivity varies widely and is subject to considerable estimation and upscaling uncertainty at field scales. In summary, the chosen facies-proportion and mean-$K$ groups represent plausible depositional/parameter-uncertainty end members for fluvial–alluvial plains and are intended to provide decision-relevant bounding behavior rather than posterior probabilistic estimates. To enhance the practical application guidance value of this study, we added a description of parameter value considerations in the Section 3.2.1 and Section 3.2.2, and provided the corresponding environmental scenarios.

Revised information is in Line 367-373 in the tracking version manuscript: From a sedimentological perspective, in fluvial–alluvial systems the areal proportion of coarse deposits (e.g., gravel/sand bodies produced in paleochannel zones) versus floodplain fine deposits can vary substantially at the basin scale, reflecting the coupled effects of stream power and sediment supply, channel migration, floodplain aggradation and development (Bridge, 2009). Accordingly, Group A (near-equal proportions) represents a more mixed and interbedded architecture consistent with frequent channel migration and facies switching, whereas Group B (fine-dominated mixtures) represents a low-energy and/or distal floodplain setting where fine deposits are more prevalent and coarse bodies are more isolated.

Revised information is in Line 415-423 in the tracking version manuscript: As is well known, $K$ varies widely and is subject to considerable estimation and upscaling uncertainty at field scales. To isolate the role of individual lithofacies, three model groups were designed in which only the mean $K$ of a single lithofacies was increased

threefold, while the other two remained unchanged. The choice to expand by three times also takes into account the uncertainty of $K$ at a medium to high level. In Group 1, the mean $K$ of GCS was raised to 138.06 m/d. In Group 2, the mean $K$ of MFS was increased to 31.02 m/d and in Group 3, the mean $K$ of SC was increased to 0.36 m/d. In all cases, the variance of $K$ was preserved, and the underlying heterogeneous sedimentary architecture remained unchanged. Thus, any changes in dispersion can be attributed to altered interfacies $K$ contrast and the resulting redistribution of flow among facies.

(7)In the Results section, please focus on "presentation + brief explanation". Lengthy discussions about cross-scale or literature comparison can be systematically elaborated in the Discussion section to avoid repetition and redundancy of the text.

**Reply:** Thank you for this comment. We have optimized the description in the Results Section and focused on key outputs and concise explanations. Other extended cross-scale interpretations and literature comparisons have been moved to the Discussion Section.

(8)There are several grammatical issues in the text. For example, line 93, "less than < 1‰" should be modified to "< 1‰";Line 175,"Scale-II" is recommended to be consistently referred to as "Scale II"; line 320,"…dispersivity shows a power…" is recommended to modified as "dispersivity exhibits a power-law increase with time.". Therefore, it is recommended that the authors carefully revise and polish the English writing throughout the manuscript.

**Reply:** Thank you for this comment. We have carefully revised and polish the English writing throughout the manuscript to solve such mistakes.

(9)In Figure 3, it is recommended to also mark key information such as the location of the Nen River and its upstream and downstream relationships.

**Reply:** We agree and have updated Figure 3.

[Figure]

**Figure 3. An example of multiscale model: (a) 3-D facies, (b) 2-D sections; and Scale II model: (c) 3-D facies, (d) 2-D sections. Section lines at *x* = 600m, 3000m, 7600m, 11000 m and *y* = 8000m, 12000m, 16000m, 18000m**

(10)In Figure 2, it is suggested that "phreatic water aquifer" in the title should be changed to "phreatic aquifer" directly.

**Reply:** Thank you for this comment. We have modified the title as "**Schematic diagram of lithofacies composition of the layered structural model of Qiqihar phreatic**  **aquifer**".

(11)Regarding flow field calibration (in Figure 5), it is currently stated that "The simulated water levels show good agreement... closely following the 1:1 line.". although the trend looks good on the graph, but specific values such as RMSE, NRMSE, and R² are missing.

**Reply:** Thank you for this comment. Quantitative metrics are indeed necessary. We will add REMS to support the statement of good agreement.

 The simulated water levels show good agreement with the observed values, closely following the 1:1 line, which proves. This visual consistency is supported by a relatively small error (RMSE = 0.507m), indicating that the water flow model constructed in this study reliably capturesreproduced the groundwater dynamics of the study area.

---

## Author Comment (AC5)

**Response to comments from Anonymous Referee #2**

**Overview of Anonymous Referee #2:**

This study combined borehole soil sampling, groundwater level data, and groundwater flow and transport modeling to examine the impact of subsurface heterogeneity on dispersion at the basin-scale. Overall, the dataset and modeling framework are promising, and the study has the potential to make a valuable contribution once the issues below are addressed.

**Reply:** We appreciate the reviewer's general impression on the value of our manuscript. We have updated the manuscript considering his/her valuable feedback. Below we addressed all the reviewer's comments and suggestions.

**Major comments:**

(1)The novelty and knowledge gap can be better articulated. In particular, the current introduction may not set the stage for this study well. Consequently, the three research objectives lack literature review support and strong motivations.

- For instance, the reviewer would be curious what motivated the author to perform an uncertainty analysis. Was it because existing literature shows a wide range of different observations in the field? Was it because a new model was used and the model involved some uncertain parameters whose uncertainty is unknown? Was it because there has been an inconsistency between existing observations and models? Or something else?

- Another example is why did they want to quantify the relative contributions of sedimentary architectural attributes, hydraulic statistics, and source size? There seems to be no well-organized discussions on the existing knowledge about how these three factors control the dispersion processes under field-representative aquifers and large-scale flow fields.

- A similar question arises for the uncertain proportion part.

**Reply:** We appreciate the reviewer's request for a clearer articulation of the novelty, the knowledge gap, and the motivations behind our research objectives—particularly the uncertainty analysis and the attribution of controlling factors. The uncertainty analysis in our study is motivated by two closely related considerations: (i) basin-scale

transport remains strongly non-ergodic for long times, thus requiring quantification of how differences between realization outputs change over time; and (ii) field-representative architectural and hydraulic inputs are inherently uncertain under limited site characterization, so it is important to understand how this structural/parametric uncertainty propagates to dispersion metrics and plume predictions. Actually, a series of simulation results in this study indicate that uncertainty is not just numerical noise, but rather predictable results caused by how the plume samples heterogeneous porous media. Most importantly, our targeted uncertainty/scenario analysis is based on recent comprehensive studies. In Section 3.2, we explicitly state that a recent global sensitivity analysis (Ren et al., 2023) indicated that a small set of geologically interpretable factors, most notably facies volume proportions and in-facies mean hydraulic conductivity, exerts first-order influence on non-reactive solute dispersion across regional to basin scales. We have emphasized that our uncertainty analysis is based on these findings. Sedimentary architectural attributes, hydraulic statistics, and source size represent distinct and practically important controls on basin-scale plume evolution: (1) sedimentary architecture governs connectivity and pathway structure; (2) hydraulic statistics (especially facies-conditioned mean $K$ and contrasts) govern velocity contrasts that drive solute dispersion; and (3) source size/geometry governs early-time sampling of heterogeneity and therefore controls the magnitude of near-source uncertainty and the onset of macrodispersion scaling. Our objective is to quantify their relative roles within one consistent, field-representative basin model so that the results can inform parameterization priorities and monitoring/management decisions. This objective already has been stated explicitly in the **Introduction**, **Results** and **Discussion** sections: lines 58-72; lines 333-340; lines 453-487. To further clarify this motivation, we have strengthened the Introduction and the opening paragraph of Section 3.2 by explicitly stating the purpose of our work.

(2)The writing and logical flow can be improved through substantial revisions. Please see the specific comments for some examples.

**Reply:** We appreciate this comment. We have systematically revised the manuscript according to the specific comments.

(3)The manuscript involves quite a number of jargons and unclear sentences, which may require clarification or rephrasing. Below are some examples:

- "From pre-asymptotic dynamics to uncertainty propagation"
- "geometric connectivity"
- "general influence"
- "hierarchical organization"
- "Although the general influence of heterogeneity has been extensively investigated, the scale-dependent effects of hierarchical organization, particularly under basin-scale flow conditions, remain inadequately quantified."
- "lithofacies"
- "macroform scales"
- "macro-dispersion"
- "finer-scale heterogeneity"
- "Non-ergodic"
- "basin-scale structure"
- "macroscopic spreading and dispersion across scales"
- "A total of 57 boreholes … were collected"
- "To bias-correct parameter estimates affected by incomplete exposure of sections, …"

**Reply:** We appreciate these detailed comments. In fact, many of the terms exemplified here are specialized in hydrogeology. Providing extensive supplementary explanations might compromise the readability of the entire text. However, based on this comment, In the revised manuscript, we have strived to reduce the use of equivocal technical terms and improve clarity in the revised manuscript.

**Specific comments:**

(1)What is the relationship between "finer-scale", "micro-scale", "meso-scale", "macro-scale", "macroform scales", "large-scale", "laboratory-scale", "facies-scale" "field-scale", "region-scale", "kilometer-scale", "basin-scale"? Without clear definitions or quantitative descriptions (like "kilometer" did), it is hard to understand the specific scales.

**Reply:** Thank you for this important comment. We agree that our manuscript used multiple "scale" descriptors, but some of which are commonly used in the literature and

were not defined quantitatively in the current version. Since this study did not conduct full-scale simulations, from the micro-scale to the macroform scale, and analyses based on one research site, we only adopted the multiscale sedimentary architecture framework to characterize the relatively coarsened and refined heterogeneity. These correspond to large-scale (Scale II) and small-scale (Scale I) measurements. Detailed scale divisions can be found in the literature by Scheibe and Freyberg (1995) and Gelhar et al. (1992). To provide quantitative anchors, we have added some explanatory statements to avoid misunderstandings as much as possible. The red content is the modified information, and the black content represents the original text (the same applies below).

Revised information is in Line 142-145 in the tracking version manuscript: Under this classification, Scale I and Scale II are used in a relative sense to denote two levels of heterogeneity representation within the same aquifer: Scale I resolves finer lithofacies variability, whereas Scale II represents a coarsened description in which Scale I facies are aggregated into composite units that preserve the dominant architectural organization.

(2)Line 30: Has the "laboratory- and sandbox-scale study" been done in this work? If not, this argument can be misleading.

**Reply:** Thank you for this comment. Those laboratory- and sandbox-scale investigations were conducted in separate, previously published studies by our research team, using sediments and boundary conditions from the same field site. These studies are cited here to place the current watershed-scale findings within a coherent multi-scale research framework.

(3)Line 30: The reason why the current study can support studying groundwater management at data-limited regions is not trivial? Please briefly explain the reasoning herein or in the main text.

**Reply:** Thank you for this comment. Actually, our intent is not to claim that data-limited regions can be managed without any uncertainty; rather, we show how a minimal set of

measurable descriptors can be used to generate defensible uncertainty bounds on basin-scale plume metrics, which is directly relevant for management tasks such as screening-level risk assessment, monitoring-network design, and prioritization of additional characterization. As stated in the manuscript, our framework clarifies (i) which heterogeneity information must be retained for reliable regional/basin-scale transport prediction and (ii) how predictive uncertainty persists and should be accounted for. Specifically, we show that for regional-scale prediction it is sufficient to preserve "dominant geological contrasts, lithofacies proportions", these parameters are comparatively observable or inferable from sparse boreholes/cross-sections and basic hydraulic characterization, while finer architectural details are most critical near a release source or at early transport stages. Consistent with this, we emphasize that "reliable regional forecasts require not only calibrating mean hydraulic properties but also retaining the geological contrasts and connectivity that govern transport, together with explicit recognition of scale-dependent averaging effects". From a groundwater-management perspective, this translates into actionable guidance for data-limited settings: prioritize characterization of contrasts/connectivity proxies and facies proportions, and interpret plume forecasts through uncertainty bounds that reflect persistent non-ergodicity rather than assuming rapid convergence to a single effective dispersivity.

(4)Line 60: It is unclear how the arguments "Recent work …" and "High-resolution …" logically connects with each, as well as how they connect with the prior discussions. That said, it is hard to catch the key message behind these arguments.

**Reply:** Thank you for this comment. We have reorganized the Introduction section to more clearly illustrate the connections with the previous studies.

Revised information is in Line 47-69 in the tracking version manuscript: Scheibe and Freyberg (1995) systematically introduced the concept of hierarchical organization from sedimentology into hydrogeology to characterize aquifer heterogeneity. They pointed out that aquifer heterogeneity is not a single-scale problem but rather results from the nested and superimposed geological units at different hierarchical levels (e.g.,

micro-, meso-, and macro-facies). In practice, although stochastic theories and numerical upscaling frameworks have provided valuable tools to connect flow and mixing processes across spatial and temporal domains, many studies continue to compress heterogeneity into effective parameters (Dagan, 1984; Gelhar, 1992; Dentz et al., 2011). Such simplification obscures the mechanistic role of multiscale sedimentary architecture in shaping flow pathways and controlling plume dynamics, and can bias spreading and mixing-relevant predictions (Fitts, 1996; Neuman and Tartakovsky, 2009; Dentz et al. 2023; Lester et al. 2016; Yin et al., 2023).

To elucidate the mechanisms underlying scale dependent transport in the heterogeneous porous media, a series of Lagrangian-based models has been developed within the hierarchical-architecture framework. By characterizing the spatial organization of lithofacies across multiple scales, these models systematically integrate the multiscale heterogeneities into a unified representation, quantitatively linking sedimentary attributes (such as lithofacies volume proportion, mean length, and statistics of hydraulic conductivity) to transport metrics (such as dispersion and mixing). They have been successfully tested in laboratory and site-scale experiments (Soltanian et al., 2015b; Dai et al., 2020; Jia et al., 2023; Ma et al., 2025).

Revised information is in Line 95-103 in the tracking version manuscript: Furthermore, as mentioned above, existing laboratory and site-scale studies have established mechanistic links between facies geometry/connectivity and solute dispersion (Dai et

al., 2005; Ramanathan et al., 2010; Soltanian et al., 2015a). Recent global sensitivity analyses further demonstrate that limited aquifer structure parameters and hydraulic conductivity statistics exert first-order control on kilometer-scale dispersion (Ren et al., 2023). These findings enable comprehensive comparisons and discussions at experimental, field, and basin scales within the same research framework, thereby constructing a complete multiscale chain to identify which hierarchical sedimentary architecture and driving factors govern macroscopic spreading and dispersion across scales.

(5)Line 65: There was no discussion on solute transport at other scales previously. It is hard to see the connection between the prior paragraph to this statement. "Most validations have been limited to laboratory or site-scale studies." cannot be justified by the current introduction. Also, it's unclear what point the authors would like to make.

**Reply:** This description here corresponds to the sentence in the last paragraph: They have been successfully tested in laboratory and site-scale experiments (Soltanian et al., 2015b; Dai et al., 2020; Jia et al., 2023; Ma et al., 2025).

(6)Line 70: How can "kilometer-scale" be considered as "high-resolution"? What resolution can be considered as "high"?

**Reply:** Thank you for this comment. Here "kilometer-scale" describes the spatial extent of the modeled system (or simulation domain), whereas "high-resolution" refers to the level of heterogeneity representation (i.e., whether lithofacies/hydrostratigraphy is explicitly resolved versus parameterized as effective properties). This sentence was not to imply that kilometer-scale is inherently "high-resolution," but that recent studies conducted over kilometer- to basin-scale extents increasingly adopt finer-resolution, architecture-resolved representations compared with earlier effective-parameter models.

(7)Line 75: "late-time tails …" This can be good motivation for this work, while the argument "Even with these advances, ..." makes the gap ambitious.

**Reply:** We appreciate this suggestion. We have retained the "late-time tails…" discussion as a key motivation, and have revised the subsequent "Even with these advances…" sentence to state a more specific and appropriately scoped gap, focusing on basin-scale, architecture-resolved settings and uncertainty propagation and avoiding overly ambitious generalizations.

Revised information is in Line 86-89 in the tracking version manuscript: Even with these advances, architecture-resolved quantification of regional/basin-scale plume evolution under field-representative heterogeneity and boundary-driven gradients remains limited, particularly regarding the persistence of asymptotic behaviour and its uncertainty.

(8)Line 80: As suggested in my first major comments, the current introduction didn't provide a comprehensive literature review and clear research trajectory to motivate the objectives.

**Reply:** Thank you for this comment. We have strengthened the **Introduction** section to illustrate the connections with the previous studies and to state the purpose of our work clearly.

(9)"Part II the" is missing a verb.

**Reply:** Thank you for this comment. Revised information is in Line 103 in the tracking version manuscript: Section 2 introduces the geographic background of the study area, borehole data, and sedimentary architecture analysis methods;

(10)Section 2 - Method: Each subsection of the method section includes both methodology (i.e., general approach) and specific model setup (i.e., parameters and data), which sometimes can affect readability. I would suggest separating them into a methodology section and a parameterization (or data collection and model setup) section. I believe this reorganization will substantially improve the clarity and quality of the paper.

**Reply:** Thank you for this comment. We attempted this adjustment and found that the logical flow of the text would create new confusion. Because this study inevitably requires providing information on lithofacies classification and its related parameters before proceeding to the water flow and solute transport models. Presenting the text in a conventional modular format of modeling and parameters would be even more confusing, so we maintained the current structure.

(11)Line 90: "The Nen River defines …" Why is this matter? "The regional terrain is gently …" Is there any support from literature or the collected data? if so, I would either cite the references or clarify the data support.

**Reply:** Thank you for this comment. The Nen River is generalized as the boundary of the study area, which determines the simulation domain of the model. Regarding the claim of gentle terrain, we derive this information from topographic contour lines, and the regional geological survey report also concludes this. Since a valid citation format cannot be provided, this paper only provides a brief overview.

(12)Figure 1 caption: More details should be provided about the selections of boreholes and cross-sections, as well as the rational or purposes of such decisions. Additionally, what were the cross-sections used for? Or what kind of information was collected from the cross-sections?

**Reply:** We simply collected as much borehole information as possible within the area and applied it to the modeling work; no special selection was performed. The purpose of using the cross-sections and the processing results is described in detail in Section 2.2.1.

(13)Figure 2: The figure is hard to interpret without providing more details about the meanings of the acronyms and the plotted regions. In the right figures, how can one interpret the figure? Are they referring to the horizontal or vertical view of the aquifer, or something else? How are the right figures connected to Figure 1b.

**Reply:** Thank you for this comment. Since this study defines multiple lithofacies names at two scales, it would be extremely redundant to introduce them all again in the figure title. The definitions of the abbreviations are already detailed in section 2.2.1. The figure on the right side is a schematic diagram of an actual lithofacies structure, demonstrating how lithofacies at a small scale are merged into lithofacies at a large scale. It can be a vertical or a horizontalview, both of which are possible in reality. The figures on the right have no relation to Figure 1b.

(14)Line 130: Can you elaborate what does it mean by "parameterized by lithofacies volume proportions and mean lengths." or provide a brief introduction about the parameterization?

**Reply:** This study used geostatistical methods based on Markov chains to construct heterogeneous lithofacies models, and the most important modeling parameters are lithofacies volume proportions and mean length, which are the information introduced in the latter half of this paragraph and the next paragraph.

Revised information is in Line 160-161 in the tracking version manuscript: The lithofacies volume proportions ($P$) define the stationary occurrence probabilities of each lithofacies, while the directional mean lengths ($L_X$, $L_Y$, $L_Z$) quantify facies correlation scales along the principal directions.

(15)Line 130: What is the "bias-correct parameter? What does it mean by "To bias-correct parameter estimates affected by incomplete exposure of sections,"?

**Reply:** Partially exposed sections can lead to errors in the statistical calculation of the actual mean length of lithofacies. For example, the mean length is calculated by dividing the sum of consecutive lengths by the number of occurrences. Because of partial exposure, the total length may be underestimated.

(16)Line 135: How were P_S and P_D defined?

**Reply:** Thank you for this comment. We have added a definition of volume ratio in line 160 in the tracking version manuscript: "The lithofacies volume proportions (*P*) define the stationary occurrence probabilities of each lithofacies…". Also, we have added a schematic figure in the Supplementary materials (Figure S17) to demonstrate the physical meaning of volume proportions.

(17)Table 1: It is hard to understand the physical meanings of the parameters and how they were derived. I would suggest providing a schematic or conceptual plot and label them in the plot at least for some of them (e.g., Lx, Ly, Lz), while providing the formula to compute P.

**Reply:** Thank you for this comment. We have added a schematic figure in the Supplementary materials (Figure S17) to illustrate how to calculate the mean length and volume proportions.

(18)Line 155: I guess you were referring to the characteristic diameters of the grains. Please clarify. What is USBR?

**Reply:** Thank you for this comment. we indeed refer to characteristic diameters of the grain and have changed the expression in the line 193 in the tracking version manuscript: the *de* is the particle diameter corresponding to *e*% finer on the cumulative grain-size distribution curve. USBR refers to the U.S. Bureau of Reclamation grain-size based empirical approach for estimating saturated hydraulic conductivity K from characteristic diameters (e.g., $d_{10}$) and related gradation descriptors. This is a widely used abbreviation definition.

(19)Table 2: The unit should be ln(m/d). Is simga dimensionless? If not, the unit should be provided.

**Reply:** Thank you for this comment. Using (m/d) as the unit indicates that only the value of *K* has been logarithmically transformed, so the unit remains unchanged. Currently, many researches in the field of groundwater treated the variance of *lnK* as dimensionless.

(20)Line 175: The authors are recommended to justify that 50 realizations are statistically grounded by adding relevant references.

**Reply:** Thank you for this comment. We have added a sentence in line 215 in the tracking version manuscript: "This number of realizations is sufficient to obtain stable ensemble statistics (Zhou et al., 2018; Henri et al., 2020)".

(21)Figure 3 provides a more clear explanation of part of the methodology. Based on that, I would suggest using some plots (e.g., a and c) to create a conceptual figure to describe the entire methodology and adding a short subsection to introduce the conceptualization at the very beginning of Section 2 Method. This will make the rest easier to follow.

**Reply:** Thank you for this comment. We have added the corresponding content based on the aforementioned suggestions.

(22)Figure 4. What is the blue rectangle representing? Are the boundary conditions applied only to the lines or the faces?

**Reply:** Thank you for this comment. The blue rectangle represents the pollution source. The boundary conditions only applied to the lines because the river does not completely cut through the phreatic aquifer. As we mentioned in line 236 in the tracking version manuscript: "The bottom of the phreatic aquifer is an aquitard with overflow discharge, so it was set as the flux boundary".

(23)Line 225: I guess planar source was indicated in Figure 4, while point source was not. Additionally, the authors should explain the rationale of simulating the two different sources, regarding the real-world processes they are representing.

**Reply:** Thank you for this comment. Yes, the planar source was indicated in Figure 4, while point source was not. The point source and planar source are intended as two representations of contaminant loading geometry in regional aquifers. The point source scenario represents localized releases such as leakage or spills from industrial facilities, tanks, pipelines, or localized injections and discharges, where the initial plume samples

only a small portion of the heterogeneous medium. The planar-source scenario represents an extended source zone that is effectively continuous along one horizontal direction and intersects multiple flow paths at early times rather than a single localized release. These two scenarios are used not to reproduce a specific historical release, but to quantify how source dimensions affect early time sampling of heterogeneity and the resulting uncertainty in plume metrics under basin-scale flow in this research. We have added some sentences to link these two idealized source geometries to the corresponding real world loading processes and to our study objective of isolating architecture-driven controls and uncertainty propagation at basin scale.

Revised information is in Line 268-271 in the tracking version manuscript: The point source represents localized releases (e.g., spills, leaks or point discharges), whereas the planar source represents an extended source zone that intersects multiple flow paths. These two scenarios are used to quantify how source dimensions control early time sampling of heterogeneity and uncertainty in plume metrics.

(24)Line 250: I would suggest reporting the error or correlation between model and observation. Additionally, in addition to the fitness between the averaged simulated water levels and measured ones, reporting the errors that reflect the fitness of simulated water levels for each realization or their statistical values (e.g., mean and standard deviation of the error of each realization) would better verify the robustness of the model results.

**Reply:** Thank you for this comment. Quantitative metrics are indeed necessary. We have added REMS to support the statement of good agreement.

Revised information is in Line 296-298 in the tracking version manuscript: The simulated water levels show good agreement with the observed values, closely following the 1:1 line. This visual consistency is supported by a relatively small error (RMSE = 0.507m), indicating that the water flow model reproduced the groundwater dynamics of the study area.

(25)Line 280: Add references to the Borden site results. Why does higher mean velocity accelerate stability? Was it due to stronger mechanical dispersion, or something else?

**Reply:** Thank you for this comment. We have added the reference in line 332 in the tracking version manuscript. The faster stabilization observed at Borden relative to our basin-scale case is interpreted as a sampling/ergodicity-timescale effect: stabilization requires that the plume traverse enough heterogeneity structure. A higher mean velocity increases the advective travel distance per unit time, so the plume samples more of the heterogeneous velocity field in the same elapsed time, and the approach toward quasi-ergodic behavior occurs sooner on the time axis.

(26)Line 305: The sensitivity and uncertainty methods were not introduced in section 2. Please provide the details about the specific methods and the analyzed factors and parameters.

**Reply:** Thank you for this comment. The sensitivity analysis in this study leans more towards scenario comparison. Therefore, we did not specifically introduce the sensitivity analysis method, but introduced the relevant content in Section 3.2 about the background and parameter settings. We also introduced the hydrogeological representativeness of different scenarios.

Revised information is in Line 367-373 in the tracking version manuscript: From a sedimentological perspective, in fluvial–alluvial systems the areal proportion of coarse deposits (e.g., gravel/sand bodies produced in paleochannel zones) versus floodplain fine deposits can vary substantially at the basin scale, reflecting the coupled effects of stream power and sediment supply, channel migration, floodplain aggradation and development (Bridge, 2009). Accordingly, Group A (near-equal proportions) represents a more mixed and interbedded architecture consistent with frequent channel migration and facies switching, whereas Group B (fine-dominated mixtures) represents a low-energy and/or distal floodplain setting where fine deposits are more prevalent and coarse bodies are more isolated.

Revised information is in Line 415-423 in the tracking version manuscript: As is well known, $K$ varies widely and is subject to considerable estimation and upscaling uncertainty at field scales. To isolate the role of individual lithofacies, three model groups were designed in which only the mean $K$ of a single lithofacies was increased threefold, while the other two remained unchanged. The choice to expand by three times also takes into account the uncertainty of $K$ at a medium to high level. In Group 1, the mean $K$ of GCS was raised to 138.06 m/d, with MFS and SC fixed at 10.34 m/d and 0.12 m/d, respectively. In Group 2, the mean $K$ of MFS was increased to 31.02 m/d and in Group 3, the mean $K$ of SC was increased to 0.36 m/d. In all cases, the variance of $K$ was preserved, and the underlying heterogeneous sedimentary architecture remained unchanged. Thus, all solute transport simulations were carried out within the same structural framework. Thus, any changes in dispersion can be attributed to altered inter-facies $K$ contrast and the resulting redistribution of flow among facies.

(27)Figure 7 caption: Please clarify what each model was representing.

**Reply:** Thank you for this comment. Please see the reply of the question 26.

(28)Line 350: What does scale-dependent tread mean? How can realization-to-realization variability reveal the "scale-dependent tread"? What is the underlying rational and theoretical basis?

**Reply:** Thank you for this comment. In this manuscript, "scale-dependent" denotes how the magnitude of realization-to-realization variability (uncertainty) changes with the effective sampling scale of heterogeneity. Realization-to-realization variability reveals this trend because it provides a direct measure of how strongly plume metrics depend on the particular stochastic architecture realization. In our figures, this is quantified by the 10–90% confidence interval envelopes computed from 50 stochastic realizations. For example, under planar-source conditions the uncertainty envelopes are markedly narrower than under point-source conditions, reflecting a larger initial support area that samples more of the heterogeneous velocity field, like the source-area

enlargement effect, and thereby reduces realization dependence. This point is discussed in detail in the **Discussion** section.

(29)Line 355: It seems Scale II was used for these analyses. Can you comment on whether the impact of varying K values on flow and solute transport will remain similar between Scale I and Scale II models?

**Reply:** Thank you for this comment. In the uncertainty/scenario analyses (Section 3.2), we intentionally used Scale II models under planar-source release to ensure stable and consistent comparisons and to minimize realization noise; this is stated in the first paragraph of Section 3.2. Regarding whether the effects of changing $K$ are similar in Scale I and Scale II representations: our results indicate that, for basin-scale transport metrics, the qualitative response to $K$ perturbations is expected to be similar in the multiscale and Scale II models, because the dominant controls are the conductivity contrasts and connectivity of the larger-scale architectural elements. This is supported by the explicit cross-scale comparison in Figure 6, where the multiscale model and the Scale II model show close agreement in plume migration and macrodispersivity, particularly under planar-source release, demonstrating that basin-scale dispersion characteristics are effectively captured once the geometry of the controlling lithofacies at larger scales is represented.

(30)Line 415: What is the reason for this difference? What is the implication for characterizations and simulations of contamination transport at different sites? I will probably talk about the key finding of these studies and then make the comparisons with the Borden site in the second last paragraph of this section given that there were several comparisons (similarity vs difference).

**Reply:** Thank you for this comment. We have reorganized the content of the **Discussion** section and then conducted a comprehensive comparison and analysis.

(31)Line numbers should be appended to each line for the reviewer to pinpoint their comments.

**Reply:** Thanks. We use the journal's standard submission template.

---

## Author Comment (AC6)

**Response to comments from Anonymous Referee #3**

**Overview of Anonymous Referee #3:**

Very good research on aquifer heterogeneities in geological porous media. Please, find my suggestions to improve the manuscript.

**Reply:** We sincerely thank the reviewer for the positive assessment of our work and for the constructive suggestions. We have carefully considered all comments and revise the manuscript and figures accordingly.

**Specific comments:**

(1)Lines 35-87. Any link between hierarchical sedimentary architecture and sequence stratigraphy?

**Reply:** We appreciate this important suggestion. We agree that hierarchical sedimentary architecture is conceptually related to sequence stratigraphy in that both describe the hierarchical organization of depositional heterogeneity across scales. However, to some extent, sequence stratigraphy usually focuses on bounding surfaces and stacking patterns (e.g., channel-belt migration, avulsion-driven packages, and floodplain aggradation) that can organize architectural elements; while our workflow focuses on architecture quantified from borehole-constrained facies transition probability and mean-length statistics, rather than explicitly mapping sequence-bounding surfaces. In this study, we adopt a two-level hierarchical representation (Scale I and Scale II) following established hydrogeologic applications of hierarchical architecture. Our goal is to quantify how such hierarchical architectural organization, regardless of whether it is interpreted through sequence stratigraphy or other depositional frameworks, controls basin-scale dispersion and uncertainty propagation. To make the explanation clearer, we added relevant explanations in lines 50-51 in the tracking version manuscript: Conceptually, this hierarchical-architecture view is consistent with sequence stratigraphy, which also organizes depositional heterogeneity in a nested manner through stratigraphic surfaces and stacking patterns.

(2) Lines 37-40. "Decades of field investigations…scale-dependent transport parameters". Insert recent research that discusses the link between parameterization of heterogeneous aquifers and plume migration:

- Agbotui, P.Y., Firouzbehi, F., Medici, G. 2025. Review of effective porosity in sandstone aquifers: insights for representation of contaminant transport. Sustainability, 17(14), 6469.

- Tellam, J.H. and Barker, R.D., 2006. Towards prediction of saturated-zone pollutant movement in groundwaters in fractured permeable-matrix aquifers: the case of the UK Permo-Triassic sandstones. https://doi.org/10.1144/GSL.SP.2006.263.01.01.

**Reply:** We agree these references strengthen the motivation for architecture-aware parameterization and its implications for plume migration. We have added both references accordingly.

(3) Lines 90-114. You need to insert more detail on the sedimentological nature of your deposits. Fluvial? Which kind of system?

**Reply:** We appreciate this specific comments. Based on the borehole and cross-sections used in this study, the phreatic aquifer is composed of thick Quaternary unconsolidated deposits associated with the Nen River corridor and adjacent floodplain. Accordingly, we interpret the system as a fluvial-alluvial depositional setting. In the revised manuscript, to enhance the practical guidance of uncertainty analysis, we have added some additional information regarding the sedimentary characteristics of the study area. Added information is in Line 367-373 in the tracking version manuscript: From a sedimentological perspective, in fluvial–alluvial systems the areal proportion of coarse deposits (e.g., gravel/sand bodies produced in paleochannel zones) versus floodplain fine deposits can vary substantially at the basin scale, reflecting the coupled effects of stream power and sediment supply, channel migration, floodplain aggradation and development (Bridge, 2009). Accordingly, Group A (near-equal proportions) represents a more mixed and interbedded architecture consistent with frequent channel migration and facies switching, whereas Group B (fine-dominated mixtures) represents a low-energy and/or distal floodplain setting where fine deposits are more prevalent and coarse bodies are more isolated.

(4) Line 120. "lithofacies-types". Please, provide detail on the sedimentary environment.

**Reply:** Thank you. We have provided brief sedimentological interpretations for each lithofacies class (Scale I) and for the aggregated units (Scale II). Obviously, such a description is not sufficient to fully explain the sedimentary environment. We have insert some sentences after the lithofacies definitions to describe depositional meaning and environment.

Added information is in Line 150-154 in the tracking version manuscript: From a sedimentological perspective, these sediments are interpreted as a river-alluvial-floodplain system associated with the Nen River. Coarse gravel/sand bodies represent channel zone (or paleochannel) deposits, while fine sand containing clay-silt lenses represents floodplain and overflow deposits. Statistical analysis of borehole lithofacies characteristics also revealed a lateral grain size reduction towards downstream from the river, and a prevalent vertical structure of coarse sand overlaying fine sand in the aquifer.

(5) Line 203. Very well-known equation that we teach to MSc students. I would delete it.

**Reply:** We agree. We have removed the explicit equation from the main manuscript.

**Figures and tables**

Figure 1. Make letters on the map larger.

**Reply:** Unfortunately, when we tried to enlarge the letters on the map, some parts overlapped, especially the nail symbol for the borehole was covered up. So, we just keep it as it is for now.

Figure 1. Increase the graphic resolution for all the three images.

**Reply:** We agree. We have increase the resolution.

[Figure]

**Figure 1. (a) Geographic location map of the study area; (b) Boreholes and cross-sections**

Figure 4. Two general head boundary and one specified flux boundary? Please, improve the figure.

**Reply:** Thank you for this comment. However, the boundaries shown in Figure 4 correspond exactly to the generalized content in the text.

Figure 5a. Make also here letters on the map larger.

**Reply:** We agree. We have modified the figure.

[Figure]

**Figure 5. (a) Distribution of groundwater level measurement points in 2020; (b) Fitting diagram between simulated water level and measured data**

Figure 5a. Increase the graphic resolution for this map.

**Reply:** We agree and change have made.

Figure 5b. Report complete error statistics (e.g., ME, MAE, RMSE etc etc) on the graph.

**Reply:** Thank you for this comment. Quantitative metrics are indeed necessary. We have added REMS to support the statement of good agreement.

Revised information is in Line 296-298 in the tracking version manuscript: The simulated water levels show good agreement with the observed values, closely following the 1:1 line. This visual consistency is supported by a relatively small error (RMSE = 0.507m), indicating that the water flow model reproduced the groundwater dynamics of the study area.